# Environmental Performance of Nitrogen Recovery from Reject Water of Sewage Sludge Treatment Based on Life Cycle Assessment

Ali Saud [1,*] , Jouni Havukainen [1] , Petteri Peltola [2] and Mika Horttanainen [1]

[1] Department of Sustainability Science, LUT University, FI-53851 Lappeenranta, Finland
[2] Department of Energy Technology, LUT University, FI-53851 Lappeenranta, Finland
* Correspondence: ali.saud@lut.fi

**Abstract:** Recovering and recycling nitrogen available in waste streams would reduce the demand for conventional fossil-based fertilizers and contribute toward food security. Based on life cycle assessment (LCA), this study aimed to evaluate the environmental performance of nitrogen recovery for fertilizer purposes from sewage sludge treatment in a municipal wastewater treatment plant (WWTP). Utilizing either air stripping or pyrolysis-derived biochar adsorbent, nitrogen was recovered from ammonium-rich reject streams generated during mechanical dewatering and thermal drying of anaerobically digested sewage sludge. A wide range of results was obtained between different scenarios and different impact categories. Biochar-based nitrogen recovery showed the lowest global warming potential with net negative GHG (greenhouse gas) emissions of $-22.5$ kt $CO_2$,eq/FU (functional unit). Ammonia capture through air stripping caused a total GHG emission of 2 kt $CO_2$,eq/FU; while in the base case scenario without nitrogen recovery, a slightly lower GHG emission of 0.2 kt $CO_2$,eq/FU was obtained. This study contributes an analysis promoting the multifunctional nature of wastewater systems with integrated resource recovery for potential environmental and health benefits.

**Keywords:** adsorption of ammonia; biochar; Life Cycle Assessment (LCA); nitrogen recovery; sewage sludge; waste to energy

## 1. Introduction

Resource conservation and recovery have gained immense attention globally in the past few years. The concerns related to the growing population, excessive extraction and utilization of raw materials, irresponsible consumption, and scarcity of basic materials have gathered researchers and scientists to find solutions that benefit humans and help to mitigate the environmental burden. As a crucial component in the urban sewage system, wastewater treatment plants (WWTPs) extract organic and inorganic pollutants that would otherwise leak into the environment and create potential hazards to the ecosystem and human health. The ever-increasing demand for wastewater treatment also increases emissions and generates piles of sewage sludge [1], which, on the other hand, emphasizes the need to recover and reuse the resources available in wastewater. Global nutrient needs and waste-to-energy potential among the main drivers; future WWTPs as "ecologically sustainable" technological systems are expected to strengthen the energy–nutrient–water nexus and, thus, become an integral part of the circular economy [2].

Nutrient recycling from WWTPs reduces the demand for conventional fossil-based fertilizers and contributes toward food security. Nitrogen is the most limiting nutrient to crop production [3], yet its recovery from wastewater treatment has become a research focus only recently. Nitrogen fertilizers are manufactured through the energy-intensive Haber–Bosch process using natural gas, from atmospheric nitrogen to plant-available ammonium nitrogen (NH$_4$-N). Contributing up to 2% of global energy consumption and causing significant greenhouse gas (GHG) emissions, the extensive production of mineral

nitrogen for fertilizers via chemical synthesis has raised economic and environmental concerns. Globally, food production can utilize only 17% of the applied nitrogen fertilizer, while the rest is lost to water bodies and the atmosphere [4]. Moreover, part of the elemental nitrogen transformed into mineral fertilizer ends up in human waste in the form of urea and ammonium, and municipal WWTPs are required to remove this nitrogen to avoid eutrophication. The European Union (EU) Urban Waste Water Directive (91/271/EEC), established to prevent adverse effects of wastewater discharge into natural water streams, specifies a minimum reduction of 80% for phosphorus and 70–80% for nitrogen [5]. However, albeit widely employed in modern WWTPs, nitrogen removal brings no additional benefits besides complying with effluent concentration limits [4]. Instead, recovering nitrogen would allow better utilization of anthropogenic nitrogen sources while saving energy and raw materials.

Inlet wastewater streams are characterized by high volumes but low concentrations; hence, without a concentration step, they are too dilute for profitable resource recovery [4]. Nitrogen, nevertheless, accumulates in the activated sludge generated in the wastewater treatment process. The nitrogen is subsequently released back to the aqueous phase during anaerobic digestion, a method commonly used for sludge stabilization, and when the anaerobically digested sludge is mechanically dewatered for further processing, a nitrogen-rich liquid fraction (reject water) with $NH_4$-N concentrations of up to 1.5 g/L is formed [6,7]. Reject water is the most nitrogen-rich stream at a WWTP, containing 15–25% of the total nitrogen content, but less than 5% of the total volume of the influent wastewater [8]. Furthermore, the remaining solid fraction (sewage sludge) carries a notable amount of nitrogen, up to 8% (dry basis), among other major plant nutrients [9]. Targeting both these waste streams for nitrogen recovery would improve the total recovery rate and allow maximal utilization of the nitrogen sources available in WWTPs [10].

Handling excess sewage sludge produced during wastewater treatment is a common problem worldwide. In the EU alone, the amount of sewage sludge has increased enormously, by around 70% from 6.5 Mt to 10.9 Mt of dry matter during 1992–2015 [11]. Currently, sewage sludge is disposed of and reused in different ways in EU member countries, including landfilling (6%), composting and other applications (12%), agricultural use (35%), and incineration (37%) [12]. Because of increasingly stringent legislation, limited space available in landfills, and soaring environmental and health issues due to the presence of harmful contaminants, e.g., heavy metals, microplastics, pharmaceutical waste, pesticides, and substances found in personal care and household products, traditional methods such as landfilling and agricultural application after biological treatment are considered problematic [13,14]. Instead, thermochemical conversion processes, e.g., pyrolysis, gasification, and incineration, have attracted significant attention as an alternative route for sludge disposal. Via thermal processing, the quantity and toxicity of sewage sludge can be reduced with simultaneous recovery of the embedded energy and chemical assets [15,16]. While requiring advanced equipment and operations, the thermochemical conversion could provide superior economic performance, efficiency, and volume reduction compared to competing sludge management technologies [17].

Pyrolysis, a thermal degradation process under inert or anoxic conditions at moderate to high temperatures (300–700 °C), converts different types of sewage sludge (raw, digested, and waste-activated) into products with added value [18]. The process results in the production of liquid pyrolytic oil (bio-oil), solid biochar, and non-condensable gases (syngas) [19]. Bio-oil is considered a potential source of energy that can fuel boilers, combustion engines, and turbines. Alternatively, bio-oil can be upgraded and refined for specialty chemicals. Biochar has shown potential benefits as a phosphorous-rich soil amendment, a carbon-neutral fuel, a low-cost adsorbent, and a replacement for carbon black, among other applications promoting environmental remediation [20]. The yield and properties of pyrolysis products depend on several factors, such as temperature, residence time, pressure, and feedstock composition. Even after mechanical dewatering, sewage sludge contains a substantial amount of moisture (73–84%). Pyrolyzing wet sewage sludge will generate a

steam-rich atmosphere inside the reactor; consequently, the liquid product will be more diluted, and the amount of non-condensable gases will increase [13]. To avoid complexities and improve pyrolysis performance, the water content in sludge can be reduced to 5–10% via thermal pre-drying. During thermal drying, a considerable proportion of the nitrogen is released within the drying fumes, which, after condensation, results in a nitrogen-rich liquid stream (condensate) that can be directed to nitrogen recovery together with the reject water from mechanical dewatering [10].

The concepts of sustainability, resource recovery, and climate change mitigation have developed a growing interest in the modelling of sewage sludge treatment systems. Besides technoeconomic assessments, life cycle analysis (LCA) has a pronounced role in selecting suitable sludge management strategies in terms of different spatial and temporal scales [21]. LCA aims to quantify the environmental performance of a process or a system by accounting for the emissions associated with material and energy flows throughout the entire life cycle. Recently, Ding et al. [22] reviewed the progress in LCA research performed on sewage sludge management and compared the environmental sustainability of existing and emerging technologies with the purpose of nutrient recovery and energy saving. Lam et al. [23] provided a summary of 65 LCA studies with different methodological practices and different scopes of nutrient removal/recovery. Clearly, the focus has been on various strategies for phosphorous recovery, while few studies only have concentrated solely on nitrogen. Kar et al. [24] and van Zelm et al. [25] examined the life cycle environmental impact of nitrogen recycling from WWTPs, considering air stripping to recover ammonia from side streams generated during sludge dewatering. Despite the different conditions and assumptions, both studies showed overall environmental benefits of the integrated removal, recovery, and fertilizer production over ammonia removal-only systems.

This study aimed to evaluate the environmental performance of nitrogen recovery for fertilizer purposes from sewage sludge treatment in a municipal WWTP. Three different scenarios, one without and two with nitrogen recovery, were investigated and compared in terms of nitrogen recovery rate and potential environmental impacts. The base case scenario without nitrogen recovery included anaerobic digestion as a conventional method to stabilize raw sludge. Since biological sludge processing alone is insufficient to remove harmful substances, pyrolysis with thermal pre-drying was considered as a post-treatment method for sludge disposal. Utilizing either air stripping or pyrolysis-derived biochar adsorbent for nitrogen recovery, the reject water generated during mechanical dewatering and the condensate generated during thermal drying were targeted as a combined source of nitrogen.

## 2. Results

The results for the studied scenarios and selected impact categories (climate change with and without biogenic carbon, terrestrial acidification, and marine and freshwater eutrophication) are summarized in Figure 1. Figure 2 illustrates the relative contribution of the processes to the produced and avoided emissions. Moreover, the results are compiled and presented in detail in Tables S5 and S6 (Supplementary Material).

Concerning environmental performance, definitive conclusions of superiority cannot be straightforwardly drawn. Scenario S3 (AdBC) performs better in three of the five impact categories, including climate change with biogenic carbon, freshwater eutrophication, and marine water eutrophication, whereas S1 (CWWTP) shows the lowest net impact for the remaining two categories, namely, climate change without biogenic carbon and acidification. Scenario S2 (S&S) remains in the middle, except for yielding the highest net impact in two categories: climate change with biogenic carbon and freshwater eutrophication.

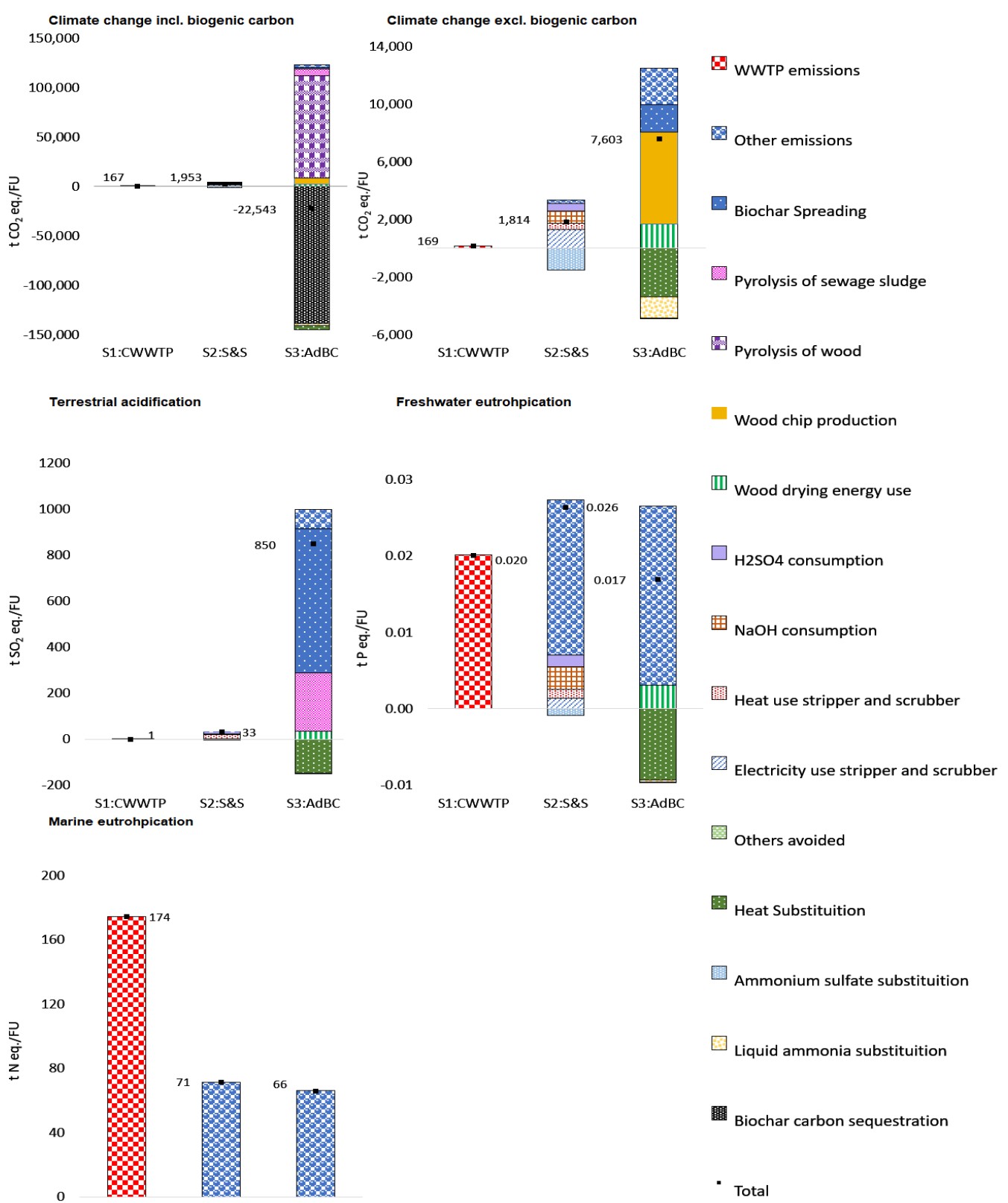

**Figure 1.** LCIA results for the selected impact categories in S1 (CWWTP), S2 (S&S), and S3 (AdBC).

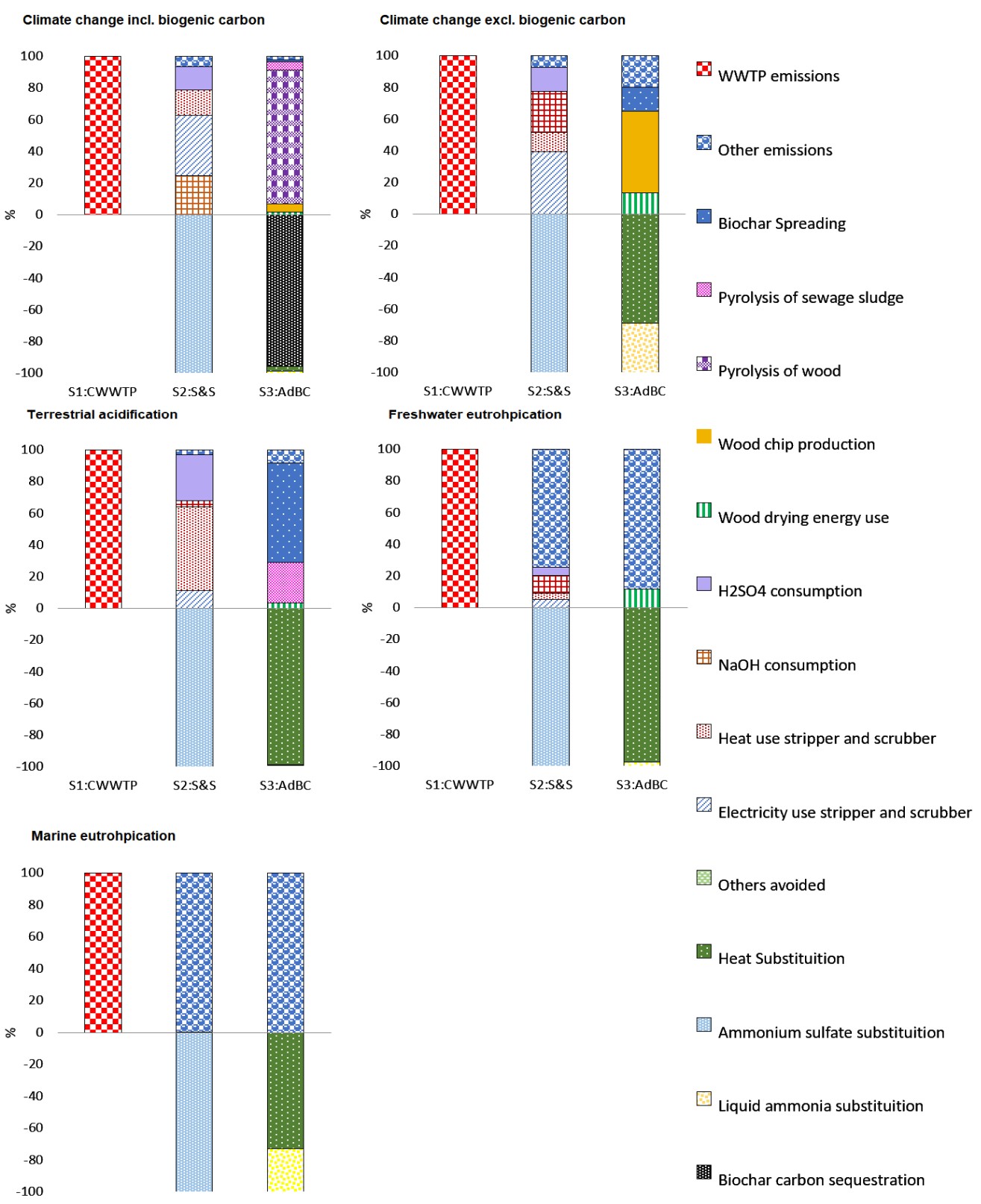

**Figure 2.** LCIA contribution assessment of scenarios S1 (CWWTP), S2 (S&S), and S3 (AdBC) for the selected impact categories.

Considering climate change with biogenic carbon, the total produced and avoided emissions from S3 (AdBC) are 120 kt $CO_2$,eq./FU and 145 kt $CO_2$,eq./FU, respectively. The

production of wood biochar yields significant emissions because wood pyrolysis requires a high amount of energy. On the other hand, 139 kt $CO_2$,eq./FU can be avoided because biochar is utilized for carbon sequestration, and a further 6 kt $CO_2$,eq./FU is avoided by substituting heat and fossil-based nitrogen fertilizers. In total, S2 (S&S) produces 3.5 t $CO_2$,eq./FU, but avoids 1.5 t $CO_2$,eq./FU. Notable emissions originate from the combined stripping and scrubbing process, which consumes electricity (1 kt $CO_2$,eq./FU) and chemicals (NaOH 0.9 kt $CO_2$,eq./FU, sulfuric acid 0.5 kt $CO_2$,eq./FU). The baseline scenario S1 (CWWTP) generates only direct emissions (0.2 kt $CO_2$,eq./FU) in this category.

In the case of climate change without biogenic carbon, the direct emissions from S3 (AdBC) are 12 kt $CO_2$,eq./FU, while a total of 5 kt $CO_2$,eq./FU is avoided. When comparing climate change with biogenic carbon, a very different net impact is obtained due to biochar-based carbon sequestration. In S2 (S&S) and S1 (CWWTP), the net impacts to the climate remain almost the same.

Table 1 shows the nitrogen recovery rate in each scenario. S3 (AdBC) with biochar adsorption yielded a nitrogen recovery rate of 540 t/a, which is 3.8% higher than that obtained via air stripping in S2 (AdBC). Nitrogen was not recovered in S1 (CWWTP).

**Table 1.** Nitrogen recovery rates.

| Scenario | Recovered Nitrogen (t/a) |
| --- | --- |
| S1 (CWWTP) | 0 |
| S2 (S&S) | 520 |
| S3 (AdBC) | 540 |

## 2.1. Contribution Analysis

The environmental impact of each process was further assessed through a contribution analysis. In the case of climate change with biogenic carbon, wood biochar production and biochar spreading cause most of the total emissions in S3 (AdBC), up to 68% and 21%, respectively (Figure 2). The high electricity demand for wood biochar production results in significant emissions. On the other hand, wood pyrolysis provides excess heat that can be used as a substitute for district heat production. However, biochar-based carbon sequestration contributes 96% of the avoided emissions, so the total impact of ammonia and heat substitution is rather limited—only 4%.

Regarding S2 (S&S), the electricity consumption in stripping and scrubbing is one of the main contributors to the climate change impact category with biogenic carbon, accounting for 38% of the total emissions. The consumption of sulfuric acid and sodium hydroxide (NaOH) contributes 15% and 25% of the total emissions, respectively. Determined by input mass flows, the electricity demands of the stripper and the scrubber are critical parameters [10]. The high input flows of reject water and condensate are the main reasons for the high electricity consumption obtained here. All the avoided emissions are due to nitrogen recovery, which enables the substitution of fossil-based fertilizers.

The main impact on freshwater and marine eutrophication in S2 (S&S) is caused by "other emissions", which include, e.g., transport of different chemicals, spreading of nitrogen fertilizer (ammonium sulfate), water consumption for acid dilution, and recycling of residual stripping liquid. Correspondingly, "other emissions" in S3 (AdBC) include transporting and spreading sewage sludge/wood biochar. Compared to S2 (S&S), S1 (CWWTP) performs better for freshwater eutrophication, which is mainly due to the consumption of NaOH and sulfuric acid in stripping and scrubbing, and, to some degree, due to fuel consumption for spreading fertilizers.

With respect to terrestrial acidification, the environmental impact of S2 (S&S) is mainly from the consumption of electricity, heat, and sulfuric acid in the stripping and scrubbing process. In S3 (AdBC), the main impact originates from biochar spreading, but also the pyrolysis of sewage sludge causes a notable impact. Furthermore, the substitution of district heat from wood pyrolysis generates 150 t $CO_2$,eq./FU of avoided emissions.

## 2.2. Sensitivity Analysis

The most important sensitivity ratios for each impact category (climate change including and excluding biogenic carbon, marine eutrophication, freshwater eutrophication, and terrestrial acidification) are shown in Table 2. Regarding S1 (CWWTP), nitrogen removal efficiency is the only parameter showing considerable variation in any impact categories, indicating high sensitivity for marine eutrophication. By increasing the nitrogen removal efficiency, the SR became negative; so, the impact on marine eutrophication will decrease because of the effective removal of nitrogen. Other parameters, such as electricity and heat demand, show only a minor influence on any impact category (|SR| < 0.7).

**Table 2.** Sensitivity ratios (SRs) for parameters in each scenario against selected impact categories. ▮ |SR| > 1 Particularly important, ▮ |SR| = 0.8–1 Important parameter, ▮ |SR| = 0.2–0.8 Slightly important, ▯ |SR| < 0.2 Minor importance.

| S1 (CWWTP) | CC incl. Biogenic | CC excl. Biogenic | FWE | ME | TA |
|---|---|---|---|---|---|
| Parameter | | | | | |
| Electricity | 0.58 | 0.58 | 0.65 | 0.00 | 0.66 |
| Heat | 0.28 | 0.28 | 0.35 | 0.00 | 0.34 |
| Lime | 0.15 | 0.15 | 0.00 | 0.00 | 0.00 |
| N removal efficiency | 0.00 | 0.00 | 0.00 | −8.49 | 0.00 |
| Electricity biogas | 0.02 | 0.02 | 0.09 | 0.00 | 0.06 |
| Heat biogas | −0.23 | −0.22 | 0.35 | 0.00 | 0.30 |
| **S2 (S&S)** | **CC incl. biogenic** | **CC excl. biogenic** | **FWE** | **ME** | **TA** |
| Parameter | | | | | |
| Electricity use | 0.71 | 0.76 | 0.05 | 0.00 | 0.12 |
| Heat use | 0.29 | 0.21 | 0.04 | 0.00 | 0.54 |
| $H_2SO_4$ use | −0.26 | −0.31 | 0.02 | 0.00 | 0.21 |
| NaOH | 0.32 | 0.32 | 0.08 | 0.00 | 0.03 |
| Water use | 0.00 | −0.01 | 0.00 | 0.00 | 0.00 |
| Stripper–Scrubber efficiency | −0.16 | −0.27 | −0.01 | −1.44 | −0.01 |
| Distance of fertilizer spreading | 0.00 | −0.01 | 0.00 | 0.00 | 0.00 |
| **S3 (AdBC)** | **CC incl. biogenic** | **CC excl. biogenic** | **FWE** | **ME** | **TA** |
| Parameter | | | | | |
| Nitrogen adsorption capacity, SS biochar | 0.12 | −0.11 | 0.03 | 0.0001 | −0.07 |
| Electricity demand, SS biochar production | 0.02 | 0.07 | 0.03 | 0.0001 | 0.01 |
| Heat demand, SS biochar production | 0.00 | 0.00 | 0.00 | 0.0000 | 0.00 |
| Electricity demand, wood biochar production | 0.00 | 0.00 | 0.00 | 0.0000 | 0.00 |
| Heat demand, wood drying | 0.00 | 0.00 | 0.00 | 0.0000 | 0.00 |
| $SO_2$ removal | 0.00 | 0.00 | 0.00 | 0.0000 | 0.29 |
| Carbon share in biochar | −0.31 | 0.00 | 0.00 | 0.0000 | 0.00 |
| Biochar nitrogen usability | 0.00 | 0.00 | 0.00 | 0.0000 | 0.00 |
| Nitrogen adsorption capacity, wood biochar | 0.52 | −0.50 | 0.13 | 0.0002 | −0.31 |
| Electricity demand wood biochar production | 0.04 | 0.12 | 0.06 | 0.0002 | 0.00 |
| Heat demand biochar | 0.05 | 0.10 | 0.13 | 0.0002 | 0.04 |
| Wood processing emissions | 0.28 | 0.84 | 0.00 | 0.0000 | 0.00 |
| Yield of wood biochar | −6.83 | −0.43 | 0.26 | 0.0004 | 0.09 |
| Excess heat production | −0.19 | −0.40 | −0.51 | −0.0009 | 0.15 |
| Substituted district heat emissions | 0.00 | 0.00 | 0.00 | 0.0000 | 0.01 |
| C share remaining in soil | −11.37 | 0.00 | 0.00 | 0.0000 | 0.00 |
| Carbon content of wood biochar | 9.57 | 0.00 | 0.00 | 0.0000 | 0.00 |
| CF (Carbon footprint) | 5.77 | −0.09 | 0.02 | 0.0000 | −0.06 |
| Steam from biomass and natural gas | 4.06 | 12.62 | −0.01 | −0.0002 | −0.04 |

CC incl. biogenic = Climate change including biogenic carbon; CC excl. biogenic = Climate change excluding biogenic carbon; **FEW** = Fresh water eutrophication; **ME** = Marine water eutrophication; **TE** = Terrestrial acidification

Indicated by the highest and the lowest SR values, the results in S2 (S&S) are most sensitive to two parameters, i.e., electricity use and stripper–scrubber efficiency. The impact on global warming potential increases by increasing the electricity demand, whereas other impact categories remain unaffected. On the other hand, the efficiency of the stripping and scrubbing process has a negative SR in the marine eutrophication impact category, so a

decreasing efficiency increases the eutrophication impact significantly (e.g., a 10% decrease increases the impact by 14%).

Among all variables in S3 (AdBC), the following parameters showed the highest influence on global warming potential: wood biochar yield, carbon share remaining in soil, total carbon in wood, CF (carbon footprint), and steam obtained from biomass and natural gas. Biochar yield with SR < −6 implies considerable variation in results; if the yield is increased, the impact on the climate change categories will decrease (e.g., a 10% increase will decrease the impact by 68%). However, the range of biochar yield is typically broad and it can be controlled by varying the conditions during pyrolysis. Thus, the conditions should be carefully set to produce a high yield of char.

Furthermore, the amount of steam obtained from biomass and natural gas combustion significantly impacts both climate change categories. Therefore, the parameter is considered extremely sensitive, because replacing a small amount of biomass with natural gas as the source of steam generation would result in significant variation in results. On the other hand, the higher the biochar yield from pyrolysis, the higher the need for external energy from biomass or natural gas. Consequently, it is necessary to pursue an optimum yield from pyrolysis. The share of carbon remaining in the soil is in turn the most sensitive parameter for the climate change category with biogenic carbon. If this variable decreases by 10%, the emissions will increase by 110%. Nevertheless, the parameter is considered highly uncertain because the long-term experimental data on carbon persistence in different soil conditions is rather limited.

## 3. Materials and Methods

The potential environmental impacts in each scenario are evaluated through a systematic LCA methodology based on ISO standards 14,040 and 14,044 [26,27]. The study includes the four steps of LCA: goal and scope definition, life cycle inventory, life cycle impact assessment (LCIA), and interpretation of results. The following impact categories were selected based on a literature review [28]: global warming potential, terrestrial acidification, marine water eutrophication, and freshwater eutrophication. Environmental performance modeling was carried out using the GaBi 10.5.1.124 software and employing the ReCiPe 2016 v1.1 (midpoint hierarchist timeframe) technique. ReCiPe indicators, which provide information on the environmental issues related to the inputs and outputs of a product system [26], are commonly utilized due to their reliability [29].

### 3.1. Goal and Scope Definition

The goal of this work is to assess the environmental performance of recovering nitrogen from sewage sludge treatment in a municipal WWTP. Nitrogen is recovered from liquid waste streams generated during mechanical dewatering and thermal drying of anaerobically digested sewage sludge. The digestate from anaerobic digestion is first dewatered and then directed to thermal drying to further reduce the water content before post-treatment via pyrolysis. The resulting water streams are rich in ammonium ($NH_4$-N) and combined for effective nitrogen recovery. Three different scenarios to manage these streams are evaluated and compared in terms of nitrogen recovery rate and potential environmental impacts:

- Scenario S1 (CWWTP) incorporates conventional treatment of reject water and condensate in a municipal WWTP; accordingly, nitrogen is not recovered but mostly removed and released into the atmosphere as $N_2$ through nitrification/denitrification.
- Scenario S2 (S&S) utilizes air stripping in combination with gas scrubbing to recover nitrogen. The two streams with recoverable nitrogen (reject water of mechanical dewatering and condensate from thermal drying) enter a stripper, and air is added in the stripper to convert ammonium to ammonia gas; subsequently, ammonia gas is absorbed in sulfuric acid in a scrubber to produce ammonium sulfate fertilizer.
- Instead of air stripping, Scenario S3 (AdBC) considers nitrogen recovery from reject water and condensate through ammonia adsorption on biochar derived from sewage

sludge and wood pyrolysis. The biochar doped with ammonia is applied to land for soil enhancement and carbon sequestration, substituting fossil-based nitrogen fertilizers.

The functional unit of this study is 870 kt/a (thousand metric tons per year) of reject water and 45 kt/a of condensate. The functional unit is based on a case study on Viikinmäki WWTP of Helsinki Region Environmental Services Authority (HSY), which was studied earlier by Havukainen et al. [21] and Saud et al. [10]. Illustrated in Figure 3, the system boundaries include the treatment of nitrogen-rich water streams (reject water and condensate).

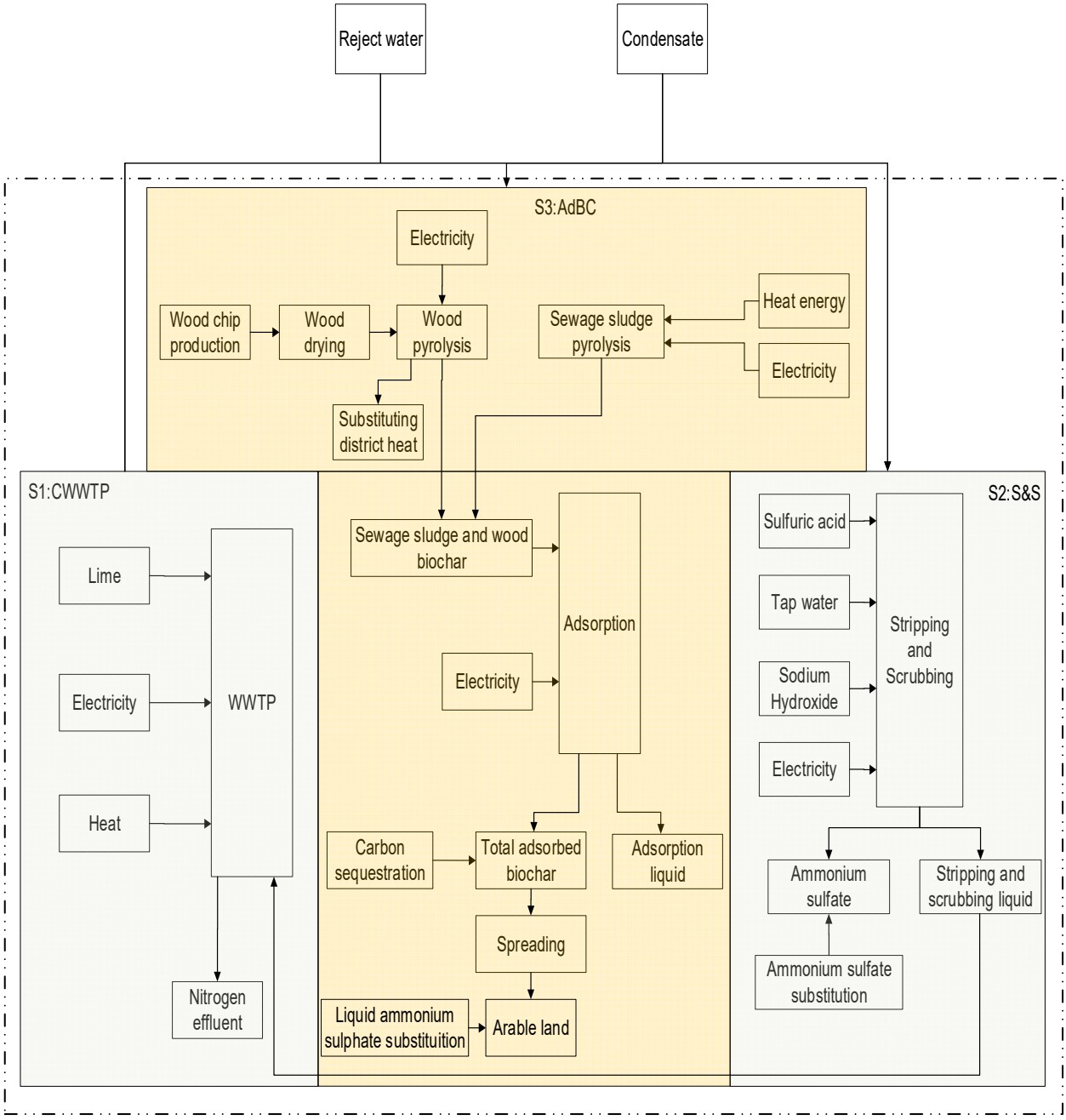

**Figure 3.** System boundaries and processes in each scenario.

### 3.2. Life Cycle Inventory
3.2.1. Reject Water and Condensate

Sewage sludge is directed to anaerobic digestion and the anaerobically digested sewage sludge (digestate) is mechanically dewatered for further processing, and an ammonium-

rich liquid stream (reject water) is formed. In the baseline configuration in Scenario S1 (CWWTP), the reject water is recirculated back to the wastewater treatment process for nitrogen removal via nitrification/denitrification. After dewatering, the digestate is directed to thermal drying to further reduce the water content. Up to 12% of the total nitrogen ($N_{tot}$) in the digestate is released within the drying fumes [30], and after condensation, another ammonium-rich liquid stream (condensate) is formed. By combining these streams, a considerable amount of nitrogen can be recovered and recycled. The properties of the reject water and the condensate are presented in Table 3.

**Table 3.** Reject water and condensate properties.

| Reject Water | | | |
|---|---|---|---|
| **Parameter** | **Value** | **Units** | **References** |
| $N_{tot}$ | 1 | kg/t | [31] |
| $NH_4$ | 0.8 | kg/t | |
| $NH_4$-N | 0.62 | kg/t | |
| **Condensate** | | | |
| **Parameter** | **Value** | **Units** | **References** |
| $N_{tot}$ | 0.09 | kg/t | [32] |
| $NH_4$-N | 0.09 | kg/t | |

### 3.2.2. Wastewater Treatment Plant (WWTP)

The WWTP considered in this work is based on data obtained from Viikinmäki WWTP in Helsinki, Finland, operated by the HSY [30]. The properties and parameters of the plant are given in Table 4.

**Table 4.** WWTP properties.

| Parameter | Value | Unit | Reference |
|---|---|---|---|
| Electricity | 1.52 | MJ/t of water | [31] |
| Heat | 1.33 | MJ/t of water | |
| Lime | 0.030 | kg/t of water | |
| N removal efficiency | 85 | % | |

The required heat for the WWTP is produced by biogas and the electricity to the WWTP is supplied partially by biogas (64%). The remainder of the electricity is assumed to be supplied either by the grid mix in Finland or renewable electricity (wind power). Table S4 (Supplementary Material) shows the power grid mix in Finland, including production and importation. A sizable portion of renewable electricity (47%) and electricity with low emissions (34.7%) are produced in Finland, and less than 20% of the whole energy mix comes from fossil fuels. Sweden accounts for most (18%) of the imported electricity, with fewer emissions than the Finnish energy production mix. Only 8% of the imported power comes from Russia, but due to the high emissions intensity of Russian electricity, it is responsible for 25% of the total emissions [33]. Since wind power in Finland has the lower emission factor compared to other renewable energy sources, it has been chosen as an example of renewable energy [34].

### 3.2.3. Stripping and Scrubbing

In S2 (S&S), a stream combined of reject water and condensate is introduced in the stripper to convert dissolved nitrogen (ammonium) to gaseous ammonia. After stripping, the nitrogen-containing air is delivered to the scrubber from the bottom, and liquid acid is either sprayed on top of or through a dense column to take in ammonia and create liquid ammonium sulfate. The inventory data of the stripping and scrubbing process is summarized in Table 5.

**Table 5.** Stripping and scrubbing parameters.

| Parameter | Value | Unit | Reference |
|---|---|---|---|
| Electricity use | 0.028 | MJ/kg | [35] |
| Heat use | 0.188 | MJ/kg | [35,36] |
| $H_2SO_4$ use | 3.5 | kg $H_2SO_4$/kg $NH_4$-N | [37] |
| NaOH | 3.3 | kg NaOH/kg $NH_4$-N | [21] |
| Water use | 2.1 | kg water/kg $NH_4$-N | Calculated |
| Stripper–Scrubber efficiency | 95 | % | [38,39] |
| **Transport** | | | |
| Biochar | 43 | km | [40] |
| Fertilizer | 43 | km | [21] |
| $H_2SO_4$ | 201 | km | [21] |

### 3.2.4. Adsorption on Biochar

In S3 (AdBC), biochar derived from sewage sludge digestate and wood pyrolysis is used as an adsorbent. The amount of biochar from sewage sludge alone is insufficient; therefore, it is necessary to pyrolyze wood to fulfill the adsorbent requirement. Before pyrolysis, wood must be chopped and dried, and electricity is required in the pyrolysis reactor. Wood drying requires electricity and heat (steam), which is produced using biomass and natural gas. In wood biochar production, excess heat originating from non-condensable pyrolysis gas is utilized for district heating and it is substituting the Finnish average district heating mix [41].

Ammonia adsorption is carried out by introducing the streams of reject water and condensate into an adsorption column filled with biochar. The energy data of sewage sludge biochar are obtained from Refs. [10,21], and the mass flow rates of sewage sludge and biochar are collected from Ref. [10]. Additional parameters, e.g., the biogenic carbon footprint of biochar recovery caused by biochar land application, were obtained from Ref. [41]. Table 6 summarizes the parameters considered in the pyrolysis process.

**Table 6.** Parameters of the adsorption system.

| Parameter | Value | Unit | Reference |
|---|---|---|---|
| **Sewage sludge biochar** | | | |
| Mass of SS biochar | 12,000 | t | [10] |
| Mass of sewage sludge | 65,000 | t | [10] |
| Nitrogen adsorption capacity | 0.004 | kg N-$NH_4$+/kg biochar | [42–44] |
| Electricity demand, SS biochar production | 0.827 | MJ/kg biochar | [21] |
| Heat demand, SS biochar production | 0.003 | MJ/kg biochar | [21] |
| Electricity demand, wood biochar production | 0.750 | MJ/kg biochar | [21] |
| Heat demand, wood drying | 0.003 | MJ/kg biochar | [21] |
| $SO_2$ removal | 0.021 | kg $CO_2$, eq./kg biochar | [21] |
| Carbon share in biochar | 34% | | [41] |
| Biochar nitrogen usability | 64% | | [45] |
| Carbon footprint biogenic | 0.45 | kg $CO_2$, eq./kg $CO_2$ | [41] |
| **Wood biochar** | | | |
| Mass of wood biochar | 97,000 | t | Calculated |
| Mass of wood | 280,000 | t | Calculated |
| Nitrogen adsorption capacity | 0.005 | kg N-$NH_4$+/kg biochar | [46–48] |
| Electricity demand wood biochar production | 0.252 | MJ/kg removed water | [41] |
| Heat demand biochar | 4.504 | MJ/kg removed water | [41] |
| Moisture (wet wood) | 28% | | [41] |
| Moisture (dry wood) | 10% | | [41] |
| Wood processing emissions | 0.018 | kg $CO_2$/kg wood | [49] |
| Yield of wood biochar | 0.34 | kg biochar/kg dry wood | [50] |
| Excess heat production | 4.9 | MJ/kg wood | [41] |
| Carbon content of wood biochar | 34% | | [41] |
| C share remaining in soil | 68% | | [41] |
| CF (carbon footprint) biochar land application | 0.45 | kg $CO_2$,eq./kg $CO_2$ | [51] |
| CF wood pyrolysis gas combustion | 0.45 | kg $CO_2$,eq./kg $CO_2$ | [51] |

*3.3. Life Cycle Impact Assessment*

The results for selected impact categories (global warming potential with and without biogenic carbon, terrestrial acidification, marine water eutrophication, and freshwater eutrophication) are analyzed through a life cycle impact assessment. First, a contribution analysis is performed to show the main contributing processes. The results of each individual process are presented discretely in terms of direct and avoided emissions. In this way, the least and the most important processes can be identified, which helps to better understand the results [52].

Second, a sensitivity analysis is used to show the most sensitive parameters. Ref. [53] suggests using sensitivity ratios (SR). SR is calculated as a ratio of two relative changes by dividing the relative change of the total result by the relative change of the individual parameter. Ref. [54] asserts that parameters with SR values greater than 0.8 are significant, and those with SR values greater than 1 are particularly significant. Parameters with SR values less than 0.2 have just a minor impact on the overall results. The usefulness of SR, however, is reliant on the effect category. Hence the SR results should be assessed inside an impact category rather than being compared between them [55]. Third, the range of the net result was determined using a "high" and "low" performance sensitivity analysis using the parameter values shown in Table S7 (Supplementary Material) for the chosen impact categories.

## 4. Discussion

A 90%/10% mixture of wood/sewage sludge biochar is generated during pyrolysis in S3 (AdBC). When applied to soil for carbon sequestration, one metric ton of this mixture removes 1.41 metric tons of $CO_2$,eq. from the atmosphere. Depending on the biochar source, this value is commonly between 0.8–2.9 t $CO_2$,eq./t [56]. According to Ref. [57] biochar derived from sewage sludge has a carbon sequestration value of 0.8 t $CO_2$,eq./t, while Ref. [58] suggest that biochar from forest residue could sequester carbon in the range of 2–2.6 t $CO_2$,eq./t.

The biochar-based carbon capture in S3 (AdBC) resulted in net negative $CO_2$ emissions. It has been estimated that the amount of carbon sequestered by biochar could increase globally to 0.3–2 Gt $CO_2$ per year by 2050. On the other hand, the yield and properties of pyrolysis products depend on several factors, such as temperature, residence time, pressure, and feedstock composition. Thus, the results obtained for S3 (AdBC) are considered highly sensitive, and further data would be needed to reliably assess the global warming potential of ammonia adsorption on biochar [59]. However, despite different methodologies and different conditions in biochar production, recent studies suggest that biochar could potentially neutralize greenhouse gas emissions while facilitating carbon capture [60].

The electricity consumption and the use of chemicals ($H_2SO_4$ and NaOH) in the stripping and scrubbing process are the main contributors to the environmental impacts in S2 (S&S). In the current work, an electricity consumption of 0.2 MJ/kg $NH_4$-N was assumed, but values as low as 0.01 MJ/kg $NH_4$-N can be found in the literature [35]. Generally, the electricity demand is determined by equipment design, operational conditions, and efficiency [61]. In addition, the environmental performance is greatly affected by the source of electricity generation—when using renewable electricity (wind power), the net emissions will reduce by 95%.

The consumption of chemicals can also be optimized. The selection of acid in the scrubbing process is determined by the requirement of the final product. Here, sulfuric acid ($H_2SO_4$) was used to produce ammonium sulfate. In addition to its use as nitrogen fertilizer in agriculture, ammonium sulfate has a wide range of potential applications. For example, it is used as a wood preservative and as a chemical in flame retardants [62]. However, as a source of key macronutrients N and S, the main target for the ammonium sulfate recovered from waste streams is the substitution of synthetic ammonium sulfate produced by the energy-intensive Haber–Bosch process. Alternatively, sulfuric acid could be replaced by organic acids such as citric acid or acetic acid to recover ammonium [63].

Furthermore, it is possible to reduce the environmental impacts by reutilizing spent sulfuric acid from petroleum refineries [64,65]. By using nitric acid ($HNO_3$), the end product would be ammonium nitrate, the most widely used nitrogen fertilizer after urea. Ammonium nitrate has been used in mining, construction, and yeast production industries. Moreover, it can be utilized as a component in insecticides or as an adsorbent for nitrogen oxide [66].

## 5. Conclusions

This study aimed to evaluate the environmental performance of nitrogen recovery for fertilizer purposes from sewage sludge treatment. Three different scenarios, one without and two with nitrogen recovery, were investigated and compared in terms of nitrogen recovery rate and potential environmental impacts. Utilizing either air stripping/scrubbing or pyrolysis-derived biochar adsorbent, nitrogen was recovered from ammonium-rich side streams generated during mechanical dewatering (reject water) and thermal drying (condensate) of anaerobically digested sewage sludge. The results show that targeting both these streams for nitrogen recovery would improve the total recovery rate and allow efficient utilization of the nitrogen sources available in WWTPs.

Scenario S3 (nitrogen recovery via ammonia adsorption) performed better in three of the five impact categories considered in this work, including climate change with biogenic carbon, freshwater eutrophication, and marine water eutrophication. The baseline scenario S1 (conventional treatment without nitrogen recovery) showed the lowest net impact for two categories, namely, climate change without biogenic carbon and acidification. In terms of environmental impacts, S2 (nitrogen recovery via air stripping and subsequent scrubbing) remained between these two scenarios. Overall, the main climate impact was caused by biochar production and utilization for carbon capture. Ammonia capture and substitution of nitrogen fertilizers appeared to cause only minor effects on climate change.

The stripping and scrubbing process for nitrogen recovery could be further improved. Specifically, the production of electricity and chemicals (NaOH, sulfuric acid) caused significant emissions. Potential improvements could be obtained by using renewable sources for electricity or replacing some of the chemicals. Furthermore, alternative process designs, e.g., steam stripping and subsequent condensation instead of air stripping and scrubbing, could provide additional benefits and reduce the net environmental impact.

The use of pyrolysis-derived biochar in ammonia recovery and further utilization for soil improvement and carbon sequestration appeared highly beneficial. While carbon sequestration is not directly related to nutrient recycling, it is shown to be relevant for sludge-based nutrient recycling pathways. However, massive quantities of biochar would be required due to the possibly low adsorption capacity, affecting the total cost of nitrogen recovery.

Solutions that simultaneously address global concerns as well as local human and ecological health are increasingly needed. This study contributes an analysis promoting the multifunctional nature of wastewater systems with integrated resource recovery for potential environmental, economic, and health benefits. The results can be used, e.g., by WWTP utilities in planning for approaches to climate change mitigation.

**Supplementary Materials:** The following supporting information can be downloaded at: https://www.mdpi.com/article/10.3390/recycling8020043/s1, Table S1: Parameters for S1:CWWTP scenario; Table S2: Parameters for S2:S&S scenario; Table S3: Parameters for S3:AdBC scenario; Table S4: Electricity grid mix shares and consumption; Table S5: LCIA results for S1:CWWTP, S2:S&S, S3:AdBC scenarios; Table S6: Contribution analysis results for S1:CWWTP, S2:S&S, S3:AdBC scenarios; Table S7: Sensitivity ratios for S1:CWWTP, S2:S&S, S3:AdBC scenarios. References [67,68] are cited in the supplementary materials.

**Author Contributions:** Conceptualization, A.S., J.H. and M.H.; methodology, A.S.; validation, A.S. and J.H.; data curation, A.S. and J.H.; writing—original draft preparation, A.S.; writing—review and editing, A.S., J.H., P.P. and M.H.; visualization, A.S.; supervision, J.H. and M.H.; project administration, M.H. All authors have read and agreed to the published version of the manuscript.

**Funding:** This research was performed as part of the NITRO project co-financed by the Academy of Finland (decision number 315051).

**Data Availability Statement:** The data presented in this study are available in supplementary material.

**Conflicts of Interest:** The authors declare no conflict of interest.

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
