# Peer review of "Environmental Performance of Nitrogen Recovery from Reject Water of Sewage Sludge Treatment Based on Life Cycle Assessment"

_recycling, doi:10.3390/recycling8020043_

Round 1
Reviewer 1 Report
Dear authors, the manuscript is of interest, but here some suggestions for you.
1) in your LCA it is not clear the background and foreground, can you provide details?
2) the quality of Fig. 1 is not good, because it is hard to read.
3) LCI is not clear. I suggest you to provide a SI in which you describe the method and procedure adopted to realise the inventory.
4) can you specify if the study is attributive or consequential?
Author Response
Hello
I have corrected the manuscript according to your comments. I am attaching word file of my response to your comments. Please see the attachment.
Thank you

Reviewer 2 Report
The paper has been well written. Some minor corrections that should be done before publication are as follows:
1. I think that the title should have the word ‘study’ or ‘analysis’ etc. after ‘Environmental Performance’.
2. Abstract: the results mentioned in the abstract should be more specific with numerical values, e.g. how much is the total recovery rate obtained or the percentage of rate increased instead of just general results like ‘improve the total recovery rate’. Other results should be revised too, e.g. the actual total cost of nitrogen recovery, etc.
3. The literature should be revised to review more past studies of LCA since currently only 1 study was reviewed (ref [22]). In addition, review of methods that have been applied by others for environmental performance study of nitrogen recovery should have been given. The limitations of those methods should have been explained to justify the method proposed by the authors using LCA. This can strengthen the novelty statement of the study.
4. N2 should be written with 2 in subscript, please revise.
5. Page 4: The contents in the paragraph before the bullet points and the information given in the bullet forms are redundant. Just maintain either 1.
6. Page 4: the bracket is not closed for the last sentence in the last line.
7. It is better if the authors can provide a diagram to illustrate the process in the WWTP so that readers can visualize where the reject water and condensate streams are in the plant. After that then the system boundaries can be shown.
8. Figure 1: What is the SS under Scenario 3? I believe it is different with the SS for Scenario 2. Please explain or define the abbreviation.
9. Table 1: what is Ntot? Needs explanation/definition.
10. Table 3: Please adjust the alignment of reference [21]. For the calculated water use, explanation should be given on how the calculation was done.
11. Table 4: explanation required for the calculated values too.
12. Section 2.3: Please explain how the contribution analysis was done.
13. The selected impact categories should be explained under Section 2.3 before the results.
14. How were the two sensitive parameters for S2 scenario observed? Since there are only 1 SR > 1 and there are many SR within the slightly important category.
15. Conclusion: the authors should highlight the importance/significance of the outcomes of the study to the industry/society/economics/environmental, etc.
Author Response
Hello
Thank you for the comments. I have corrected all the according to your comments in the manuscript. Please see the attachment for my response to your comments.
Thank you
